# Small Noncoding RNAs MTS0997 and MTS1338 Affect the Adaptation and Virulence of *Mycobacterium tuberculosis*

**Galina Shepelkova [1,\*], Vladimir Evstifeev [1], Mikhail Averbakch Jr. [1], Ilya Sivokozov [1], Atadzhan Ergeshov [1], Tatyana Azhikina [2]**  **and Vladimir Yeremeev [1,\*]**

[1]   Central Tuberculosis Research Institute, 107564 Moscow, Russia; vladimir-evstifeev@yandex.ru (V.E.); amm50@mail.ru (M.A.J.); i.sivokozov@ctri.ru (I.S.); cniit@ctri.ru (A.E.)
[2]   Shemyakin-Ovchinnikov Institute of Bioorganic Chemistry, Russian Academy of Sciences, 117997 Moscow, Russia; tatazhik@ibch.ru
[\*]   Correspondence: g.shepelkova@ctri.ru (G.S.); yeremeev56@mail.ru (V.Y.)

**Abstract:** Tuberculosis (TB) is currently the leading cause of death among bacterial infectious diseases. The spectrum of disease manifestations depends on both host immune responses and the ability of *Mycobacterium tuberculosis* to resist it. Small non-coding RNAs are known to regulate gene expression; however, their functional role in the relationship of *M. tuberculosis* with the host is poorly understood. Here, we investigated the effect of small non-coding sRNAs MTS1338 and MTS0997 on *M. tuberculosis* properties by creating knockout strains. We also assessed the effect of small non-coding RNAs on the survival of wild type and mutant mycobacteria in primary cultures of human alveolar macrophages and the virulence of these strains in a mouse infection model. Wild-type and mutants survived differentially in human alveolar macrophages. Infection of I/St mice with KO *M. tuberculosis* H37RV strains provided beneficial effects onto major TB phenotypes. We observed attenuated tuberculosis-specific inflammatory responses, including reduced cellular infiltration and decreased granuloma formation in the lungs. Infections caused by KO strains were characterized by significantly lower inflammation of mouse lung tissue and increased survival time of infected animals. Thus, the deletion of MTS0997 and MTS1338 lead to a significant decrease in the virulence of *M. tuberculosis*.

**Keywords:** tuberculosis; small non-coding RNA; macrophages; immune response

## 1. Introduction

*Mycobacterium tuberculosis* is one of the most successful human pathogens, which causes more deaths than any other single infectious agent. According to the latest World Health Organization (WHO) data, one third of the global population infected with mycobacteria is at risk of developing tuberculosis (TB) [1]. However, the mechanisms and factors responsible for transition from latent to active phase of the infection are still unknown. The range of manifestations of TB infection depends both on the characteristics of the host's immune response to the pathogen, and on the ability of mycobacteria to resist this response. After mycobacteria enter the host through the respiratory tract, both the development of active TB infection with the manifestation of clinical signs of tuberculosis, and asymptomatic "latent" TB infection with the presence of dormant *M. tuberculosis* is possible [2–4].

Inability of the immune system to protect the host from the pathogenic effects of mycobacteria depends on many factors; however, two main ones can be highlighted: immune system defects at various levels [5,6] and modulation of *M. tuberculosis* immune response that allows mycobacteria to survive, multiply, and cause a disease [7].

The main adaptation trait that accounts for the success of *M. tuberculosis* as a pathogen is its ability to maintain latent infection in the host tissues for a long time without causing an actual disease. During this chronic stage, the *M. tuberculosis* genome undergoes rearrangements that provide adaptation of mycobacteria to the hostile environment in

phagolysosomes of macrophages or in tuberculous granuloma [8]. As an intracellular pathogen, *M. tuberculosis* adapts to an unfavorable external environment and not only modulates its metabolism but also affects the metabolism of the infected host cell. Regulatory proteins, non-coding RNAs and their targets constitute complex regulatory networks that allow the pathogen to adapt its metabolism at different stages of infection. Attempts to understand the molecular basis of the genetic regulation of TB focus on identification of genes encoding proteins and regulons involved in its pathogenesis. Recently, a few studies were focused on non-coding RNAs, which are believed to significantly contribute to the regulatory networks responsible for *M. tuberculosis* adaptation and virulence.

At present, research in the field of small RNAs of *M. tuberculosis* is actively developing [9–11]. Small RNAs are reasonably attributed to the functions of gene expression regulators during adaptation of bacteria to changing environment. However, no direct experimental evidence has yet been found and no specific regulatory pathways for small RNAs to participate in TB pathogenesis have been suggested.

Previously, it has been reported that sRNAs MTS1338 and MTS0997 are expressed at high levels in *M. tuberculosis* and show significant stability in dormant bacterial cells corresponding to latent infection [12]. MTS1338 is poorly expressed during the exponential growth phase, but upon transition to the stationary phase, its expression increases significantly which indicates that MTS1338 may play a role in the formation of dormant *M. tuberculosis* cells and latent TB [12,13]. The level of MTS0997 expression also increases during transition from the exponential growth phase to the stationary phase [14].

Here, we investigated the impact of these small RNAs on the ability of *M. tuberculosis* to interact with the host organism. To this end we used mutant strains with deleted genes (MTS1338 and MTS0997) to infect human alveolar macrophages (AM) in vitro. Aerosol mouse model of infection was used to analyze the virulence of mutant strains as compared to the wild type *M. tuberculosis* H37Rv.

## 2. Materials and Methods

### 2.1. Mice

We used inbred I/StYCit (I/St) female mice (susceptible to TB infection) weighing 20–25 g at the age of 2–4 months. The strain was maintained at the breeding facility of the Central Tuberculosis Research Institute (CTRI). Mice were kept in non-sterile conditions in accordance with the Ministry of Health Decree 755 and the guidelines of the Office for the Welfare of Laboratory Animals A5502-01, with access to food and water ad libitum. All experiments on animals were implemented in accordance with the standards adopted by the CTRI and approved by the Ethics Review Committee (protocol code 010 from 15-01-2018).

### 2.2. Mutant Strains of M. tuberculosis

Mutant strains (KO, knock out) of *M. tuberculosis* with deletion of the small RNA genes MTS0997 and MTS1338 were obtained earlier by Dr. Elena G. Salina (Bach Institute of Biochemistry, Research Center of Biotechnology, Russia) using the method described in the manuscript [15]. The deletion of the genes was confirmed by a genome-wide sequence. The mutant strain data can be accessed from NCBI SRA (Bioproject ID: PRJNA701202).

### 2.3. Restoration of M. tuberculosis Virulence

During passaging in vitro, *M. tuberculosis* loses its virulence, which can be restored when mycobacteria enter the host. In order to restore the virulence of *M. tuberculosis* strains (H37RV wt, H37RV-MTS0997KO and H37RV-MTS1338KO), mice were infected with these mycobacteria. Three weeks after the infection, mice were sacrificed, spleens extracted and homogenized, and 50 µL of serial 10-fold dilutions of homogenates were spread on Dubos agar (BD Difco, Wokingham, Berkshire, UK), and incubated at 37 °C. After 21 days, rough mycobacterial colonies typical for virulent mycobacteria were selected, inoculated into liquid Dubos broth (BD Difco, Wokingham, Berkshire, UK), and incubated at 37 °C for

17 days; mycobacteria were collected by centrifugation at $1000\times g$ and transferred to PBS with 0.1% Tween 80. To determine the concentration of mycobacteria, drops of serial suspension dilutions were applied onto Petri dishes with the Dubos agar (BD Difco, Wokingham, Berkshire, UK). After 4 days of culturing, microcolonies were counted [16].

### 2.4. Preparation of M. tuberculosis Culture and Infection of Animals

We used *M. tuberculosis* H37Rv Pasteur as well as two H37Rv strains with knock-outs in small mycobacterial RNA MTS0997 (H37Rv-MTS0997KO) and MTS1338 (H37Rv-MTS1338KO). Mycobacteria were obtained in sufficient quantities and stored in aliquots at $-70\,^\circ$C.

Mice ($n$ = 13 per group) were infected intravenously with $10^6$ CFU of each mycobacterial strain per mouse. After 40 days, 3 mice in each group were sacrificed for lung tissue analysis. The remaining 10 mice were left to monitor survival.

### 2.5. Histology

To study pathological changes in the lungs, the tissue was frozen at $-60\,^\circ$C to $-20\,^\circ$C temperature gradient for 1 h in the Thermo Electron Shandon Cryotome (UK). 5–8 μm sections were air-dried, fixed in methanol, and then stained with hematoxylin and eosin.

### 2.6. Human Alveolar Macrophages (AM)

We used bronchoalveolar lavage (BAL) cells from non-infectious patients being treated at the Clinical Diagnostic Department of the CTRI. Informed consent was obtained from each patient prior to the initiation of our study. Adhesion of macrophages on Petri dishes was used for purification from non-macrophage cells. Macrophages were detached from Petri dishes with Versene solution. Cultivation of macrophages ($5 \times 10^4$ cells/well) with different numbers of mycobacteria (from $5 \times 10^4$ to $50 \times 10^4$ CFU/well) in vitro was performed in 96-well flat bottom plates in a medium RPMI 1640 containing 5% FCS, 10 mM HEPES and 2 mM L-glutamine without antibiotics. Purified AM suspension was subjected to re-adhesion in the plate wells and *M. tuberculosis* suspension was added to the established macrophage monolayer. The viability of mycobacteria in mixed cultures was assessed via the selective inclusion of 5,6–[$^3$H]–Uracil [17,18].

### 2.7. Flow Cytometry

The left lungs of each mouse were enzymatically disrupted as described earlier [19], and single-cell suspensions were analyzed by flow cytometry using fluorescein isothiocyanate–phycoerythrin–or allophycocyanin-labeled monoclonal antibodies against surface markers CD4, CD8, CD19, CD11c, CD11b, and Ly6G (Biolegend, San Diego, CA, USA).

### 2.8. Statistical Data Analysis

Statistical analysis was performed using a t-test (adjusted for multiple comparisons by Sidak-Bonferroni) with GraphPad Prism 7.0 software (GraphPad Software, San Diego, CA, USA).

Kaplan–Meier analysis was used to compare the survival. $p$ values were calculated using the Mantel–Cox test.

$p < 0.05$ was considered statistically significant.

## 3. Results

### 3.1. The Survival Rate of MTS1338 in Macrophages Is Lower Than That of H37RV wt

For bacteria–intracellular parasites, all multi-stage and diverse mechanisms of defense and pathogenesis converge in tissue macrophages. On the one hand, these cells provide the pathogen with a place for survival and reproduction, and on the other, they carry out effector functions and eliminate bacteria. Thus, at the first stage of the study, we compared

the survival of wild-type mycobacteria (H37RV wt) and mutants (H37Rv-MTS0997KO and H37Rv-MTS1338KO) in the human AM.

Human AMs were co-cultured in vitro with *M. tuberculosis* H37Rv wt, H37Rv-MTS1338KO, and H37Rv-MTS0997KO at the mycobacterium:macrophage ratio from 1:1 to 10:1 according to the method described by Shepelkova et al. [17]. Next, 72 h after the start of cell cultivation, the viability of mycobacteria was assessed via the selective inclusion of [$^3$H]-Uracil. The activity of macrophages was measured by the degree of mycobacterial growth inhibition in cultures infected with the different amount of bacteria as compared to non-infected cultures (control).

As can be seen in Figure 1, wild-type mycobacteria are characterized by more active growth during in vitro cultivation. The growth of mutant mycobacteria (H37Rv-MTS0997KO and H37Rv-MTS1338KO) slowed down in the medium adapted for cultivation of human AM. With the equal mycobacteria: macrophages ratio (1:1), macrophages almost completely inhibit the growth of H37Rv-MTS1338KO mutant bacteria ($360 \pm 67$ cpm), the background inclusion of [$^3$H]-Uracil in the macrophage culture without mycobacteria was $312 \pm 39$ cpm. On the other hand, with 10:1 ratio, macrophages could not cope with a high concentration of the three bacterial variants. Appropriate inhibition of mycobacterial growth was observed at a mycobacterium:macrophage ratio of 5:1.

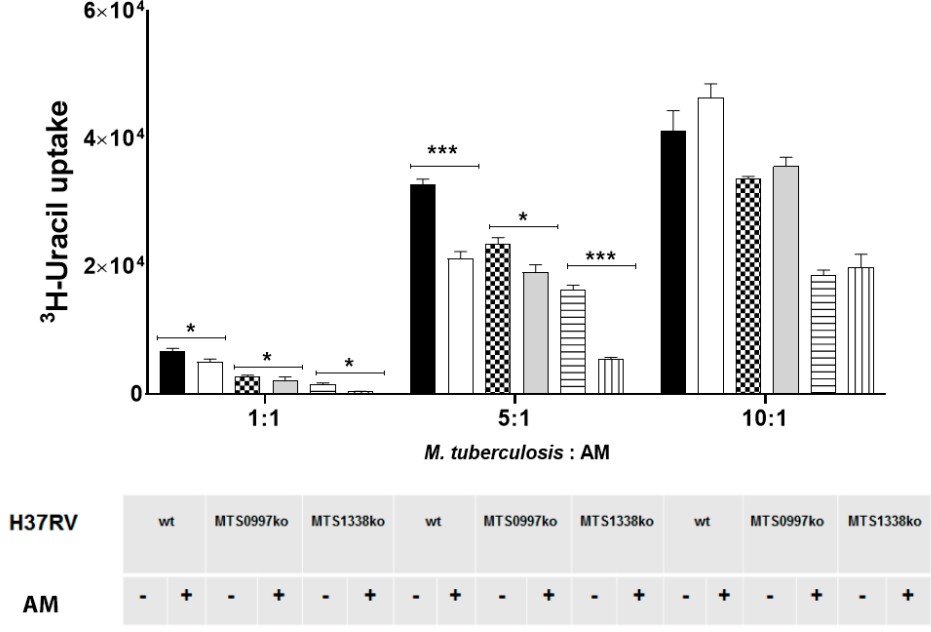

**Figure 1.** Survival of *M. tuberculosis* WT, MTS0997-KO, and MTS1338-KO strains in human AMs. Macrophages were infected with *M. tuberculosis* (TBS) strains at different ratios and cultures were analyzed for mycobacteria proliferation based on [$^3$H]-Uracil uptake both in the absence (−) and presence of macrophages ($5 \times 10^4$ cells/well) (+). Non-infected AMs were considered as background ($311.7 \pm 38.8$ cpm). The data were collected during one of three typical independent experiments (mean $\pm$ SEM, $n = 3$ per group). * $p < 0.05$, *** $p < 0.001$.

Figure 1 shows that AMs most effectively inhibit the growth of mutant mycobacteria H37Rv-MTS1338KO (compared to the growth of H37Rv-MTS1338KO bacteria without macrophages; mycobacterial growth was inhibited by 67%), while H37Rv-MTS0997KO, on the contrary, survives best in macrophages (growth of H37Rv-MTS0997KO mycobacteria in culture with AM was inhibited by 19%). In experiments with co-cultivation of *M. tuberculosis* H37Rv wt and AM at the ratio of 5 mycobacterium:1 macrophage, mycobacterial growth was inhibited by 36% (Figure 1).

### 3.2. Mycobacteria Deficient in MTS1338 and MTS0997 Have Decreased Virulence In Vivo

As a next step, we performed a comparative assessment of mutant mycobacteria and *M. tuberculosis* H37Rv wild type virulence in the in vivo system. I/St mice susceptible to TB infection were intravenously infected with three *M. tuberculosis* strains: H37Rv-wt, H37Rv-MTS1338KO, and H37Rv-MTS0997KO at a dose of $10^6$ CFU/mouse. It was 40 days after the infection that the level of inflammation and infiltration of the infected mice lung tissue was assessed. This was followed by the survival comparisons in animals infected with different variants of mycobacteria.

Analysis of lung tissue morphology performed 40 days post-infection showed that I/St mice infected with the H37Rv wt strain had severe inflammation in the lungs, as evidenced by massive cell infiltration (Figure 2A,D) and the presence of focal diffuse cell clusters around large vessels and bronchi. However, the inflammatory cell infiltration was significantly reduced in the lungs of mice infected with MTS0997-KO and MTS1338-KO strains compared to those infected with the WT strain ($p < 0.05$ and $p < 0.001$, respectively; Figure 2B–D).

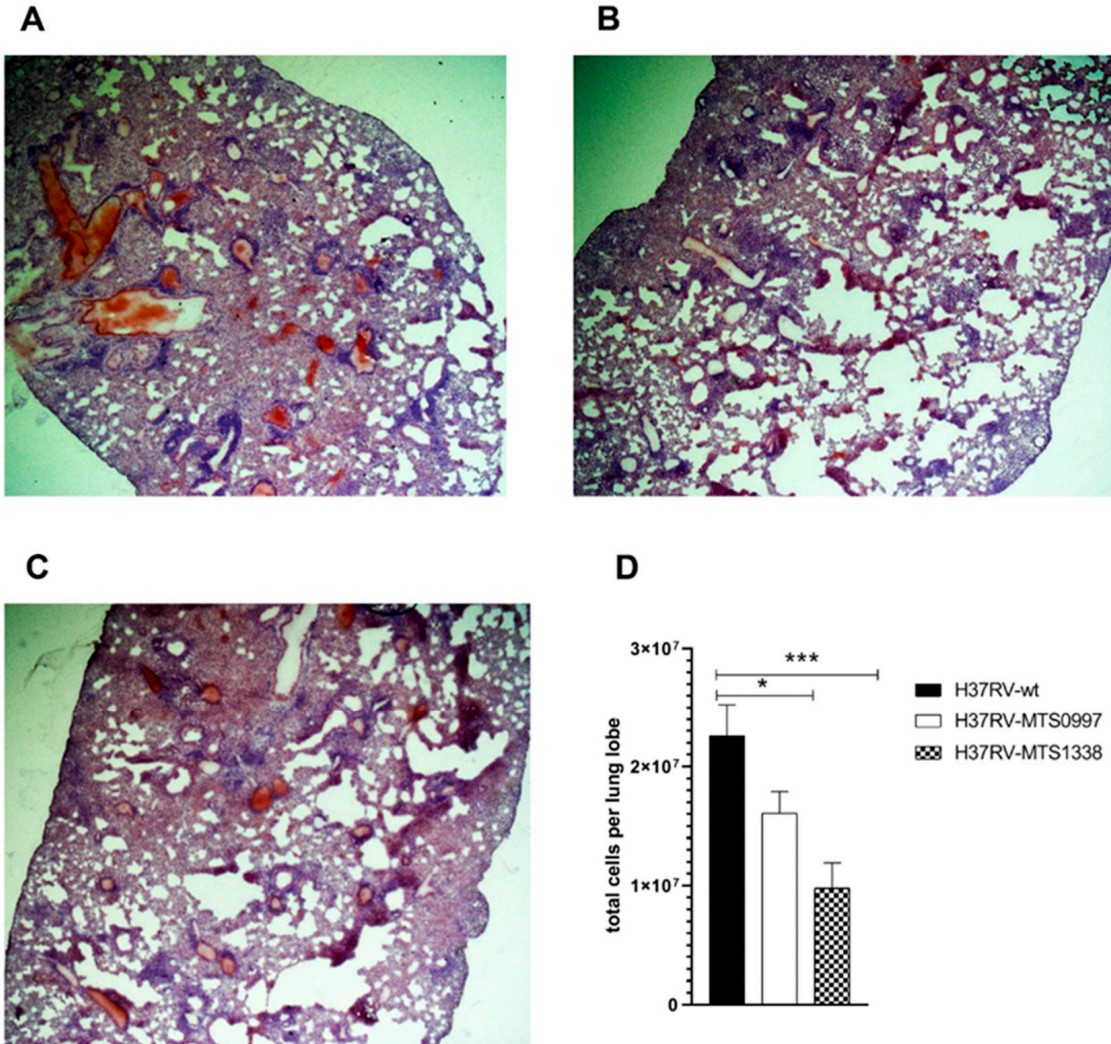

**Figure 2.** Immune cell infiltration in the lungs of I/St mice infected by the WT and mutant *M. tuberculosis*. Mice were intravenously infected with WT (**A**), MTS0997-KO (**B**), and MTS1338-KO (**C**) strains, and sacrificed 40 days post-infection. Inflammation in lung tissues was evaluated via histology in sections of the middle right lobe of the lung after staining with hematoxylin and eosin (magnification, ×25). (**D**) The total number of lung lobe derived cells from one of three similar experiments (Flow cytometry analysis) ($n = 3$ mice per group); * $p < 0.05$ (for WT and MTS0997-KO infection compared), *** $p < 0.001$ (for WT and MTS1338-KO infection compared) and ns (for MTS0997-KO and MTS0997-KO infection compared).

The developing granulomas predominantly contained epithelioid cells, large monocytes, and macrophages with hyperchromic nuclei (Figure 3). Epithelioid cells were surrounded by a shaft of monocytes and macrophages without any signs of leukocytes formed granulomas (Figure 3D). However, cell density of the inflammatory infiltrate was significantly less in the lungs of mice infected with MTS0997-KO and MTS1338-KO strains compared to those of WT strain-infected mice; macrophages and monocytes had smaller hyperchromatic nuclei and their numbers were reduced (Figure 3E,F). Comparative analysis of 5–10 slides in each group revealed only slight differences between tissues of mice infected with MTS0997-KO and MTS1338-KO strains.

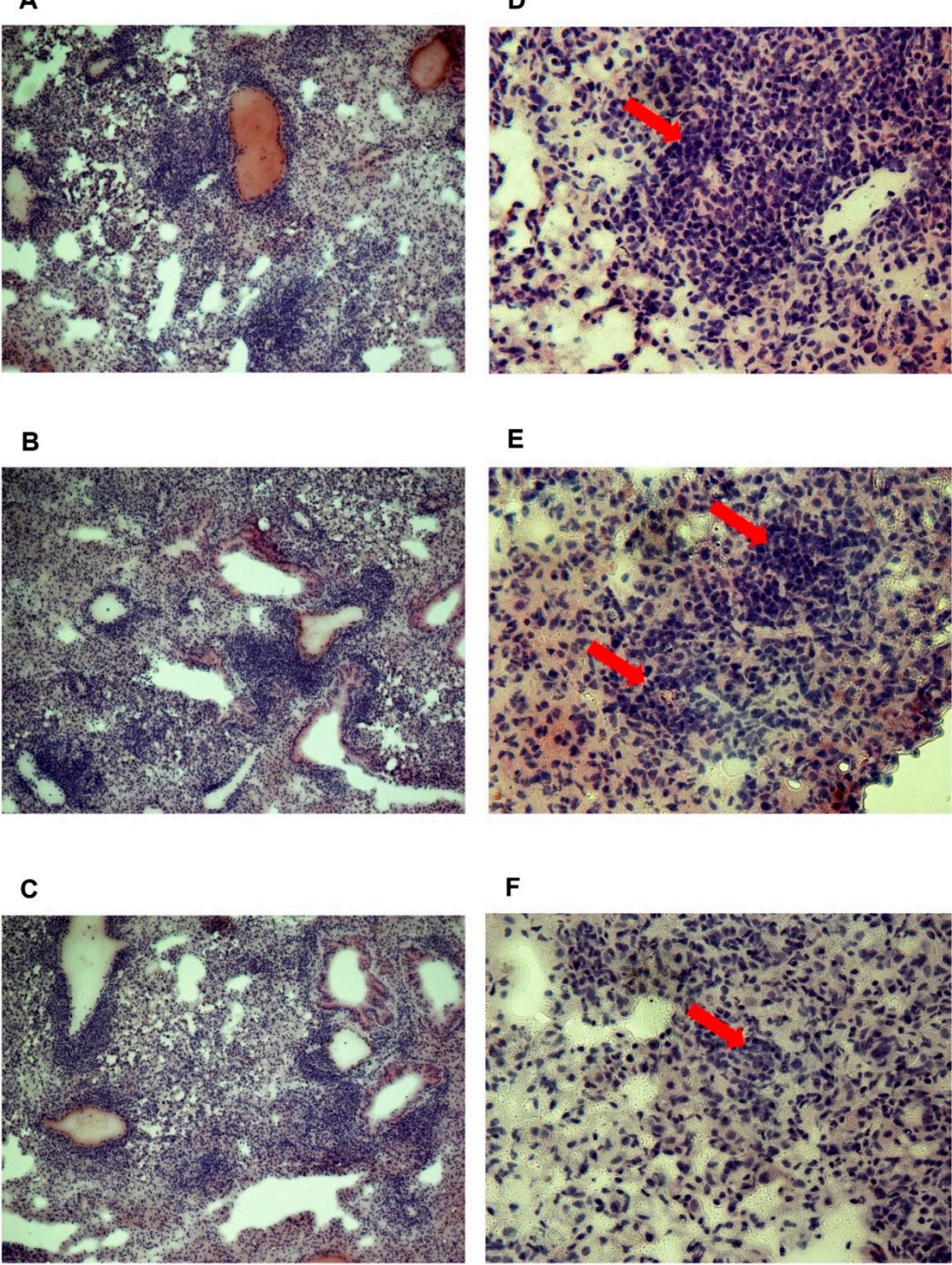

**Figure 3.** Lung tissue pathology in I/St mice 40 days after infection. Sections of the middle right lung lobe of mice infected with WT (**A**,**D**), MTS0997-KO (**B**,**E**), and MTS1338-KO (**C**,**F**) strains were stained with hematoxylin and eosin and examined for inflammation at magnification ×100 (**A**–**C**) and ×400 (**D**–**F**). Arrows indicate the sites of granuloma formation.

Consistent with the histological observations, quantification of the lymphoid cell populations infiltrating lung tissue at day 40 after infection also revealed the differences between the groups of mice infected with H37RV wt and two mutants (Figures 2D and 4).

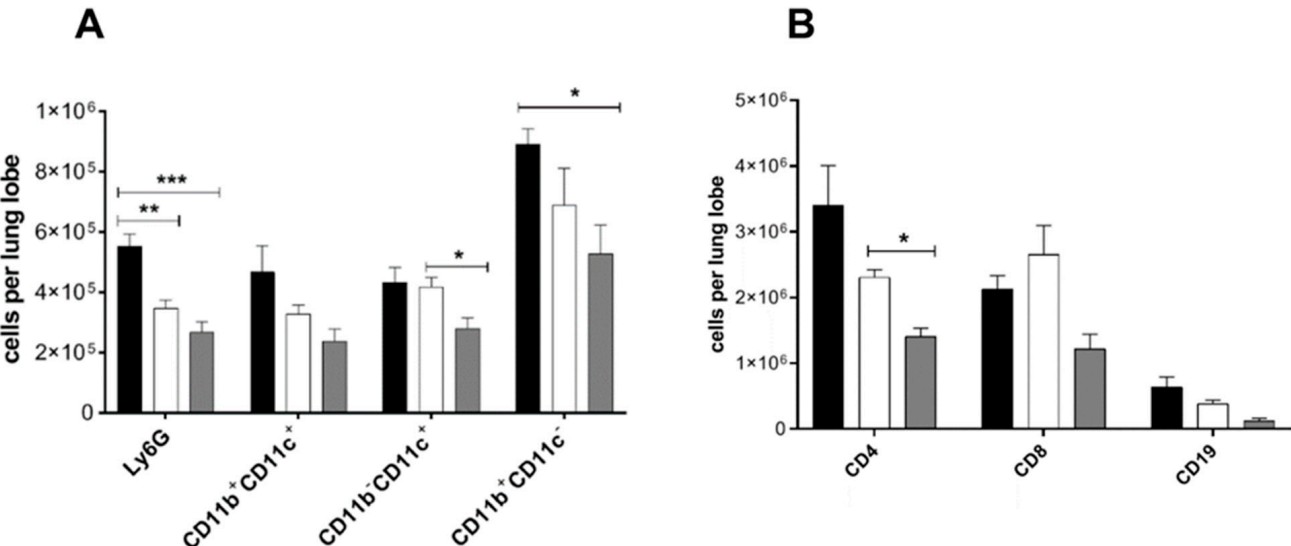

**Figure 4.** Cellular composition of the lung tissue of I/St mice infected with WT and mutant *M. tuberculosis*. Mice were infected with H37Rv-WT (black bars), H37Rv-MTS0997KO (white bars), and H37Rv-MTS1338KO (gray bars) strains, and analyzed for the presence of different populations of inflammatory cells by flow cytometry 40 days post-infection. (**A**) Neutrophils (Ly6G$^+$), tissue-resident macrophages (CD11b$^+$CD11c$^-$), and dendritic cells (CD11b$^-$CD11c$^+$ and CD11b$^+$CD11c$^+$). (**B**) Main lymphocyte subpopulations. The data are presented as the mean $\pm$ SEM of one of three similar independent experiments (*n* = 3 mice per group); * $p < 0.05$, ** $p < 0.001$, and *** $p < 0.0001$.

There was a significant decrease in the content of neutrophils (Ly6G$^+$), tissue-resident macrophages (CD11b$^+$CD11c$^-$), and the total number of lymphoid cells in the lungs of mice infected with mutant mycobacteria compared to the WT strain.

Kaplan–Meier survival analysis (Figure 5) showed that the lifespan of mice infected with mutant mycobacterial strains (H37Rv-MTS0997KO and H37Rv-MTS1338KO) was significantly longer than that in the group of animals infected with H37RV wt (H37Rv wt and H37Rv-MTS0997KO *p* = 0.01; H37Rv wt and H37Rv-MTS1338KO *p* = 0.0004; H37Rv-MTS0997KO and H37Rv-MTS1338KO *p* = 0.1–not significant).

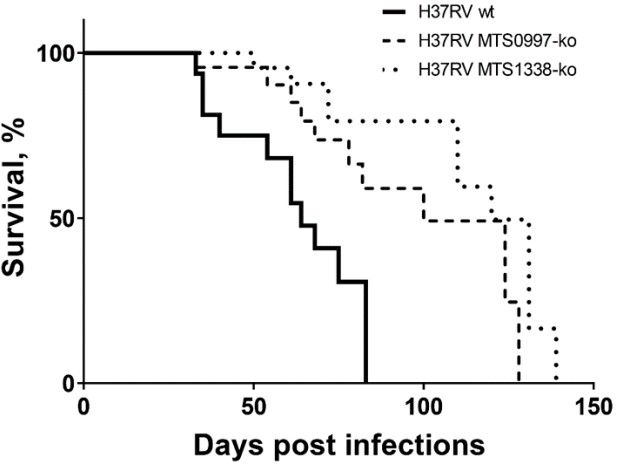

**Figure 5.** Survival of I/St mice infected with *M. tuberculosis*. WT strain, solid line; MTS0997-KO strain, dashed line; MTS1338-KO strain, dotted line. The data are representative of three independent experiments that showed similar results (*n* = 10 mice per group).

## 4. Discussion

Protection against *M. tuberculosis* infection is largely ensured by the components of the innate immune system, resident alveolar, recruited macrophages, neutrophils, and dendritic cells [19]. When mycobacteria enter the alveolar space, AMs trigger primary defense mechanisms aimed at fighting bacteria. The synthesis and secretion of pro-inflammatory cytokines and chemokines that contribute to the destruction of bacteria and/or activation of inflammatory reactions lead to the development of granulomas. However, in addition to the above, nonactivated AMs can serve as a "refuge" for the growth of mycobacteria [20–23].

AMs are located in the alveolar space with the constant exposure to both neutral antigens and pathogenic microorganisms; therefore, these macrophages must have plasticity in order to perform phagocytic functions without stimulating the development of an excessive immune response along with a rapid protective pro-inflammatory response to the pathogen.

After entering the host, *M. tuberculosis* can persist for a long time in a dormant state without any visible harm to the host and without any transmission of the infection to the contacts. To be able to persist in macrophages, mycobacterium undergoes metabolic rearrangements that provide adaptation to the aggressive environment inside a phagolysosome of a phagocyte [24]. These changes include changes in the synthesis and production of various regulatory proteins, small non-coding RNAs, and their targets. When the balance between the "parasite" and the immune system of a "host" is disturbed, *M. tuberculosis* enter a metabolically active state and starts replicating up to several billion, which leads to the destruction of host tissues and the spread of infection.

Recent studies indicate an active role of mycobacterial small non-coding RNA in this process [25–28]. To study the effect of small mycobacterial RNAs MTS0997 and MTS1338 on the ability of *M. tuberculosis* to interact with the host, we created mutants of the H37Rv strain with deleted genes of these small RNAs. Previously, it was shown that sRNA MTS1338 transcription level in the activated macrophages is significantly higher than in non-activated macrophages [25].

Small RNAs MTS1338 and MTS0997 activate mycobacterial genes involved in the bacterial persistence within a macrophage [25,27], meaning that an increase in the concentration of these small RNAs contributes to the survival of the bacteria. These observations are consistent with our studies, where AMs more prominently inhibit the growth of mutant H37Rv-MTS1338KO and also H37Rv-MTS0997KO mycobacteria as compared with wild-type H37Rv strain. Also, compared to the wild-type bacteria in an in vitro system, mutant *M. tuberculosis* demonstrate slow growth in AM culture medium. In our model, deletion of genes in mycobacterial sRNAs of MTS0997 and MTS1338 led to a decrease in bacterial virulence, which was expressed in a greater efficiency of macrophages against mutant mycobacteria compared to the wild-type H37Rv strain.

A decrease in virulence of deletion mutants of H37Rv is also demonstrated by the results obtained from TB susceptible I/St mice in an in vivo model. Previously, it was shown that transcription of small RNAs MTS0997 and MTS1338 in the mouse models of infection increases significantly with the development of an immune response to the pathogen [29,30]. In our studies, a lower degree of infiltration of the lung tissue was demonstrated in the groups of mice infected with mutant mycobacteria; the degree of specific TB inflammation was lower, as indicated by both the lower number of neutrophils and macrophages in the lung tissue (compared to the group of animals infected with wild-type bacteria), as well as the pattern of granulomas formation in the lung.

The groups of I/St mice were infected with H37Rv-MTS0997KO and H37Rv-MTS1338KO 40 days after the infection development of granulomas, while the wild-type group was characterized by the granulomas surrounded by a shaft of monocytes and macrophages. Moreover, the death of mice was postponed in the groups infected with mutant mycobacteria, i.e., infection with mutant bacteria led to a more favorable course of the infection. Thus, the deletion of MTS0997 and MTS1338 leads to a significant decrease

in the virulence and adaptation for survival in macrophages of *M. tuberculosis* at the early stages of infection.

**Author Contributions:** Conceptualization, G.S. and V.Y.; methodology, G.S., V.E., M.A.J., and I.S.; validation, V.E. and M.A.J.; formal analysis, G.S.; investigation, G.S., V.E., and I.S.; resources, I.S.; writing—original draft preparation, G.S.; writing—review and editing, G.S., V.Y., A.E., and T.A.; visualization, G.S.; supervision, V.Y.; funding acquisition, T.A. All authors have read and agreed to the published version of the manuscript.

**Funding:** This research was funded by Russian Science Foundation, grant number 18-15-00332.

**Institutional Review Board Statement:** The study was conducted according to the guidelines of the Declaration of Helsinki, and approved by the Institutional Ethics Committee of Central Tuberculosis Research Institute (protocol code 015 from 23-01-2018).

**Informed Consent Statement:** Informed consent was obtained from all subjects involved in the study.

**Data Availability Statement:** The data presented in this study are available on request from the corresponding author. The data are not publicly available due to ethical considerations.

**Conflicts of Interest:** The authors declare no conflict of interest. The funders had no role in the design of the study; in the collection, analyses, or interpretation of data; in the writing of the manuscript, or in the decision to publish the results.

## Abbreviations

TB      Tuberculosis
AM      Alveolar macrophages

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
