# Peer review of "Small Noncoding RNAs MTS0997 and MTS1338 Affect the Adaptation and Virulence of Mycobacterium tuberculosis"

_2036-7481, doi:10.3390/microbiolres12010014_

Round 1

Reviewer 1 Report

Review of « Small noncoding RNAs MTS0997 and MTS1338 affect the adaptation and virulence of M. tuberculosis »

The manuscript by Shepelkova et al. presents a study of two KOs for small RNA that were previously shown to be highly expressed in macrophages compared to exponential phase in media. These two mutant strains lacking sRNAs MTS0997 and MTS1338 were studied for their capacity to survive and replicate in macrophages and for their virulence in mice.

This is an important type of study to go further than the initial expression studies where these sRNAs have been previously described. However, the gold standard of such studies would normally include complementation of the KO strains with a vector (or integrated DNA at another locus) of the deleted gene, in these cases MTS0997 and MTS1338. If this was attempted and provided inconclusive results, it would be good to know. That being said, the authors later discuss the fact that MTS1338 has been overexpressed in a previous study (Salina et al. 2019 Front. Cell. Infect. Microbiol) and how the results obtained in that paper corroborate what is observed in the current study.

In addition to the afore mentioned, before publication in Microbiology Research we recommend many modifications to the text, and possibly a few experiments (or inclusion of results that authors likely already have). Here are the other comments:

P1 line 14, in the abstract: replace “Tuberculosis is currently the leading cause of mortality among infectious diseases” by “Tuberculosis is currently the leading cause of mortality among bacterial infectious diseases” or other changes that would take in account that COVID-19 currently has a higher death toll than tuberculosis.

Line 82, section 2.2: Re-write to for complete sentences (e.g., the first “sentence” of the section has no verb).

Line 84: “were obtained earlier by E. Salina” -> either provide a complete reference (with the details of the constructs used for knock outs) or provide the details within the manuscript.

Line 100: Not sure about the format of exponents in Microbiology Res., but I assume it should be 106, several other instances might need to be changed as well.

Line 104: add degrees to C.

Line 138: uniformize ref format ([17] instead of Shepelkova et al. 2015)

Lines 145-147: Description of the results is confusing. If the background of uracil incorporation is 312 ± 39 cpm (311.7 ± 38.8 is too much digits), then the sentence from lines 146-147 would be more clear like this:

…macrophages almost completely inhibit their growth both in the case of wild-type mycobacteria (add cpm number here) and in co-cultivation with mutant bacteria (MTS0997ko number, MTS1338ko number), the background inclusion of [3H]-uracil in the macrophage culture without mycobacteria was 312 ± 39 cpm.

Furthermore, it is not clear what the growth in absence of macrophages corresponds to, as no Mycobacteria growth media is mentioned in materials in methods with regards with this experiment. If the growth medium of AMs is used (but in absence of AMs), media is not described in materials and methods either. This also leads to an important comment on the presented results. While authors present these sRNAs as virulence factors, the only data presented on the effect of sRNA KOs in medium (rather than in macrophage or mice) is in figure 1, but this is presumably not optimal M. tuberculosis medium. In short, does the deletion of MTS0997 and MTS1338 affects growth of M. tuberculosis in general? Is the bacteria less virulent, or simply grows less thus leading to less virulence. It would to compare bacteria growth in liquid culture before comparing their growth in AMs. This way, it would demonstrate that the defect in growth is environment dependent. (Line 248-250: it is stated that compared to the wild-type bacteria in vitro system, mutant M. tuberculosis demonstrated a slow growth in AM culture medium (note that given its importance, this data should be presented more clearly, and using growth medium better adapted to M. tuberculosis would be preferable).  This could make us believe that the sRNAs are important for bacteria, but not specifically for virulence.

Figure 1: The figure is not easy to understand with the table below. Keep the same abbreviation for macrophages: AM or mph (or at least define mph in the legend if it is different than AM).

The goal of the figure is to compare the survival of the different strains of M. tuberculosis (WT, MTS0997-Ko, MTS1338-KO). The statistical analysis between the different strains should also be shown, not only within each strains between the condition with and without macrophages.

Also, correct the axis to “uracil”, not “uracile”. We also note that materials and methods does not describe radioactivity measurements to deduce intake of 3H-uracil, this sentence is used instead: “The viability of mycobacteria in mixed cultures was assessed by the selective inclusion of 5,6-[3H]-Uracil [17].” In general, it is preferable to describe (at least briefly) experimental procedures, but it is even more important in this case given that reference [17] is in Russian.

Line 168, what is mln? 106? Use appropriate denomination or international units.

Fig 2: A control with medium only and/or without treatment would be useful for comparison for the readers unfamiliar with histopathology, if the authors have such an image, they should include it. Moreover, adding arrows would help readers focus on what we are supposed to see on the tissue morphology. In D, add a bar to show significant difference between MTS0997ko and MTS1338ko, if there is no significant difference, simply add the statement (difference between MTS0997ko and MTS1338ko is not statistically significant). Also for D, presumably these are results from flow cytometry, but it is not mentioned. It is also not mentioned how infected cells are distinguished from un-infected cells in section 2.7.

Lines 184-191. Description of results from Fig 3 want to highlight a large difference between WT and KO strains. However, to this non-expert reviewer, difference in granuloma density seems to be greater between MTS0997ko (E) and MTS1338ko (F) than between WT (D) and MTS0997ko (E). More explanations might help convince non-expert readers, perhaps with arrows added to the X100 magnification as well.

Fig 3: Again, a control without any infection would be useful for comparison for non-experts.

Fig 4: adding the legend in the image directly (as in Fig2 D) would help its readability.

Line 213: Keep the same nomenclature throughout the text (MTS 0.01997KO instead of MTS0997-ko)

Line 248: What about for the mutant H37Rv-MTS0997ko?

Author Response

Dear Sir/Madame,

Thank you very much for your time taken to review our manuscript and for suggestions you made. Please find enclosed our answers to your questions.

Reviewer 2 Report

The article by Shepelkova et al. describes the effect of two small noncoding RNAs (MTS0997 and MTS1338) on the adaptation and virulence of Mycobacterium tuberculosis. The manuscript is really interesting in the topic and in the possible implications in the research field.

In the title, the strain should be complete.

Lines 18-21: This sentence is too long. 

Sometimes tuberculosis appears as TB, but not always. Please, be consistent with the nomenclature throughout the manuscript.

Introduction in interesting although some of the references are a bit old. The last paragraph (Lines 69-72) needs to be explained better. Please, improve it.

Materials and Methods section needs to be improved, particularly the section related to the mutant strains of M. tuberculosis.

Lines 82-84: Please, remodulate the sentence, it is not clear.

Line 79: ad libitum should be written in italics

It should be advisable to add the provider company of some reagents.

Figure 1 and 4 need legend to be easily understable

The number of mice used in each experiment is confused, is it 13 or just 3?

The discussion, in its first paragraphs, seems more like an introduction. It should be improved and finished with a critical statement of the results obtained. Usually, numbers of figures are not included in the discussion. Please, remove it (lines 258-261, 264-265).

Strains should be in italic in the titles of the papers that appear in the reference section.

Author Response

(The authors gave the same response as above.)

Round 2

Reviewer 1 Report

Review of second version of « Small noncoding RNAs MTS0997 and MTS1338 affect the adaptation and virulence of M. tuberculosis »

Here are some additional comments with regards to the authors modifications and comments.

#1

“Complementation studies were not in the focus of the current research.” I disagree, a complementation study would be a relevant control. That being said, it is not necessarily essential.

#2

Section 2.2. Even if the authors reference the method for mutagenesis of M. tuberculosis, it is insufficient to repeat the experiment. We would need the details of the plasmid used to make the corresponding KO. Also, while useful for those interested, providing the link of the Bioproject and NCBI SRA is not an appropriate surrogate for the plasmid sequence used. First, it does not necessarily provide the relevant information, second, it is not reasonable to ask readers to look at 7 Gbases of data to find that information. If the details of the constructs are not available because it will be part of another publication by Dr. Elena G. Salina or some other reason like that, this should at least have been mentioned in the response to reviewers. If it is the case, something should also be included in the text so that readers know where to look for (presumably another publication by Salina EG will come out with these KO), perhaps a tentative reference with “xyz et al. title, submitted)? I do not know if MDPI or Microbiology research has some rules for that.

Line 115: there is an instance of 10^4 remaining (also line 164).

#3

Figure 1 has very important results to judge the impact of the sRNA deletion with regards to growth in media or intracellular growth. Indeed, although authors still have not compared “in between strains” significance for 3H-uracile incorporation, it seems clear that at all ratios the deletion of sRNAs have impacts on growth, whether simply in media or in presence of AMs. To better present this, a table with the ratio of cpm-media / cpm-AMs could be useful (or simply add the numbers above the bars, in which case, these numbers could be compared for significance between strains).

#4

Figure 2.

Line 185: … lung lobe immune cells from…

As for the authors suggestion:

“The picture of normal mouse lung one can find for example in the “Handbook of Tuberculosis.

Immunology and Cell Biology”. 2008. Edited by Stefan H.E. Kaufmann and Warwick J. Britton.

p.248.”

I do not happen to have this in my library, I might thus end up in a situation where I can hardly judge some important results presented by the authors. If arrows cannot be used to direct the reader’s attention more precisely, perhaps some other indications, like brackets within the figure, or a very brief description to help us know what to look for (e.g. the brown/red stained patches).

#5

Regarding lines 188-195: answer from authors:

“Here we present one typical picture of each case. While serial slides were analyzed. Our statement is that inflammation is more prominent in wild-type infected lungs as compared to KO strains. The slight differences between KOs are not so important. Same as in figure 1 the arrows do not add anything for understanding.”

Then, at the end of the paragraph adding text indicating that …these are representative of many slides (if you have a number, provide it) supporting that inflammation is more prominent in wild-type infected lungs as compared to KO strains. Only slight differences were observed between KOs.

Again, having no healthy lung image to compare, and being relatively unfamiliar with histology, it is not obvious how this compares to healthy lungs.

#6

Our simple suggestion was to add the black/white/gray legend within the figure 4 (where there is plenty of space to do so) to facilitate readability, which incidentally might also have pleased reviewer 2.

#7

Line 255: Authors previously referred to figure 1, to make sure that it is clear this statement does not relate to results from the literature, simply adding “our” and “demonstrated” would help:

“Also, compared to the wild type bacteria in our in vitro system, mutant M. tuberculosis demonstrated slow growth in AM culture medium (Figure 1).”

As for the statement:

“In our model, deletion of genes in mycobacterial sRNAs of MTS0997 and MTS1338 led to a

 decrease in bacterial virulence, which was expressed in a greater efficiency of macrophages against mutant mycobacteria compared to the wild-type H37Rv strain.”

Adding the suggested ratio of survival (or growth?) “media/AM” would help support this claim, or perhaps indicate that the impact is more general and not limited to survival and growth in AMs.

Author Response

Dear colleague!

Thank you again for your efforts to improve our paper. Please, find enclosed our answers to your questions.
